# Highly Flame-Retardant and Low Heat/Smoke-Release Wood Materials: Fabrication and Properties

**DOI:** 10.3390/polym14193944

**Published:** 2022-09-21

**Authors:** Ze-Peng Deng, Teng Fu, Xin Song, Zi-Li Wang, De-Ming Guo, Yu-Zhong Wang, Fei Song

**Affiliations:** The Collaborative Innovation Center for Eco-Friendly and Fire-Safety Polymeric Materials (MoE), National Engineering Laboratory of Eco-Friendly Polymeric Materials (Sichuan), State Key Laboratory of Polymer Materials Engineering, College of Chemistry, Sichuan University, Chengdu 610064, China

**Keywords:** wood, fire-retardant coating, wood preservative, flame-retardant, smoke suppression

## Abstract

Wood is an important renewable material exhibiting excellent physical and mechanical properties, environmental friendliness, and sustainability, and has been widely applied in daily life. However, its inherent flammability and susceptibility to fungal attack greatly limit its application in many areas. Use of fire-retardant coatings and preservatives has endowed wood with improved safety performance; importantly, the cooperative effect of dual treatments on the burning behavior and flame retardancy of wood needs to be better understood. Here, a two-step treatment for wood is proposed, with a copper–boron preservative (CBP) and a fire-retardant coating. The thermal degradation and burning behavior of treated wood were investigated. The CBP formed a physical barrier on the wood surface, facilitating a charring process at high temperatures and thus suppressing the release of heat and smoke. Notably, the dual-treated wood exhibited lower heat release and reduced smoke emission compared with the mono-treated wood, indicating a cooperative effect between CBP and fire-retardant coatings, beneficial to the improvement of fire safety. This experimental work improved fire retardance and suppressed smoke release in flammable materials, and offers a new design for developing fire-retardant coatings.

## 1. Introduction

Wood is an important renewable material with excellent physical and mechanical properties, environmental friendliness, and natural beauty that has been extensively used for aesthetic, engineering, and structural applications [1,2,3]. Wood is a natural organic polymer material, highly flammable and susceptible to biological attack by termites, fungi, molds, insects, and other organisms during use, resulting in performance deterioration and functional failure [4,5]. Therefore, it is necessary to treat wood physically or chemically to improve its safety for practical applications.

Flame retardant and fireproofing treatment of wood can slow or control fire spread effectively, improving fire-safety performance. A novel flame-retardant wood was prepared through surface-impregnation on yellow birch under reduced pressure with polyelectrolyte complexes (PECs) [6]. Although the reduced pressure shortened the treatment time, the fire-retardant properties of the prepared wood were not satisfactory. Phytic acid was impregnated into the delignified wood to improve flame retardance, and the LOI of the treated wood reached 37.2% [7]. Despite the excellent flame-retardant effect, the complexity of handling limits its application. In terms of treatment application, coatings have generally been preferred over wood impregnation due to their low cost, short time consumption, and low impact on the physical and chemical properties of raw wood [8]. Currently, intumescent fire-retardant coatings are the most widely used of these materials for wood products. Researchers synthesized a series of magnesium phosphate ester flame retardants (MPEA) which they thoroughly mixed with amino resin to produce transparent intumescent fire-retardant coatings applied on wood substrates [9]. The peak heat release rate (PHRR) and total heat release (THR) of the coated wood were reduced by 40.9% and 29.0%, respectively, compared with the control wood, demonstrating a good intumescent flame-retardant properties. Furthermore, a semi-transparent dual-functional, intumescent fire-retardant self-healing water-based coating for plywood was described, which increased the fire-resistant time of coated plywood by 85% compared with pure plywood [10]. Typical intumescent fire-retardant coatings consist of three main flame-retardant additives; (1) an acid source, usually acting as a charring promoter or dehydrating agent that produces acid compounds in situ; (2) a gas source generating gas during heating to promote foaming of coatings; and (3) a char source acting as a charring agent that insulates underlying substrates and maintains the structural integrity. These components are combined with polymeric binders to form intumescent fire-retardant coatings [11]. When exposed to fire, such coatings can expand rapidly and form a stable char layer, and the carbonaceous honeycomb or porous residue acts as a barrier to heat, air, and pyrolysis products, and ultimately protects the substrate from spread of fire [12]. However, quantities of toxic fumes can be released during burning due to certain flammable components in coatings, such as epoxy compounds. Therefore, design of an intumescent fire-retardant coating system with excellent flame-retardant properties and low smoke emission remains an urgent concern.

Regarding wood preservation, chemical preservatives can efficiently resist fungal attacks and extend the service life of wood. Preservatives based on copper–chromium–arsenic (CCA), lindane, and pentachlorophenol have been reported to inhibit wood decay significantly, but these preservatives are potentially harmful to the environment [13,14]. Recently, boron-based (e.g., borax), copper-based (e.g., ammonia-soluble alkyl amine copper), and triazole-containing preservatives (e.g., propiconazole) have often been preferred for wood preservation treatments. These preservatives effectively protect wood via impregnation, and because preserved wood is widely used the effect of preservatives on the fire retardancy of wood, especially fire-retardant treated wood, cannot be ignored. Previous studies have mentioned that Cu^2+^ and borate, which correspond to the active components of certain preservatives, can strongly interact with cellulose, hemicellulose, and lignin in wood to promote the formation of aromatized structures during pyrolysis and combustion [15,16]. Therefore, some wood preservations are beneficial not only for bacterial and fungal inhibitions but also for flame retardancy in wood. However, few reports have discussed their cooperative flame-retardant and smoke-suppression effects in detail, which is an obstacle to understanding the practical situation of modified wood in fire, and to further optimization.

Here, we designed and prepared a fire retardant and wood preservative, involving two-step treatment with a CBP and an intumescent fire-retardant coating. Fire-retardant and smoke-suppression behaviors of wood before and after treatment were evaluated. To understand the contribution of preservatives and flame retardants to the burning behavior of wood, the char residues after burning different samples were studied in detail.

## 2. Materials and Methods

### 2.1. Materials

Pinus sylvestris wood was purchased from a wood factory in China; an X-1-type fire=retardant coating containing 25 wt% ammonium polyphosphate (APP), 15 wt% pentaerythritol (PER), 10 wt% melamine (MEL), and 50 wt% melamine–urea–formaldehyde resin (MUF); and a copper–boron preservative (CBP) mainly containing Cu^2+^ and borate (the mass ratio was 1:1) were prepared by the National Engineering Laboratory of Eco-Friendly Polymeric Materials (Sichuan) Lab of Sichuan University.

### 2.2. Preparation of Fireproof and Preservative Wood

As shown in Figure 1a, the wood was first soaked in a 10 wt% CBP/water solution for 24 h and then placed in a cool place to dry. After drying completely, the CBP-treated wood was painted with X-1 fire-retardant coating using a pneumatic spray gun 15 cm from its surface, and the interval between each spraying was controlled at 2 h. The above step was repeated three times until the total coating amount reached 250 g/m^2^, and then the fireproof and preservative wood were obtained after 8 h of curing. All specimens were sprayed on all surfaces except for the cabinet-method test specimens (sprayed only on the surface towards the fire). The different samples were classified as raw wood (RW), CBP-treated wood (CTW), X-1 fire retardant coating-treated wood (XTW), and CBP/X-1 fire retardant coating-treated wood (CXTW), respectively.

### 2.3. Characterization

Sample morphology was analyzed by scanning electron microscopy (SEM, Phenom Pro X, Phenom-World, Eindhoven, Netherlands). To estimate the adhesion of the coating, strength tests were performed using a universal testing machine (SANS CMT 4204, SANS, Minnesota, USA) with coated wood at room temperature. Test specimens (70 mm × 70 mm × 10 mm) were prepared according to the GB/T 14907-2002 standard. The identical coated wooden blocks were bonded with epoxy resin and immediately covered with another coated block (Figure 2a). After compressing with a 1 kg weight for three days, the samples were pulled to failure at 100 mm/min, and adhesive strength was calculated as the maximum force divided by the known overlap area. The mechanical properties testing was conducted on a universal testing machine (INSTRON 3366, INSTRON, High Wycombe, UK) equipped with a 10 kN load cell. For bending tests, samples with dimensions of 20 × 20 × 300 mm^3^ were prepared according to GB/T 1936.1-2009. The span length of the universal testing machine was set to 240 mm, and the load was applied in the tangential direction at a rate of 10 mm min^−1^. For compression testing, samples with dimensions of 30 × 20 × 20 mm^3^ were prepared according to GB/T1935-2009, and the load rate of the universal testing machine was set to 5 mm min^−1^. Thermal stability was assessed by thermogravimetry analysis (TGA, TGA 5500, TA Instruments, Delaware, USA) from 40 °C to 700 °C at a rate of 10 °C/min in a nitrogen or air atmosphere. Flame retardance was measured with a limiting oxygen index meter (LOI, HC-2C, Nanjing Jiangning Analytical Instruments, Nanjing, China) with a size of 130×6.5×3.2 mm^3^, according to ASTM D2863-97. Flame retardance was characterized using a vertical burning test (UL-94 V, CZF-3 instrument, Jiangning, China) with a sample size of 130 × 13 × 3.2 mm^3^, according to ATSM D3801-10. Burning behavior was characterized by cone calorimetry (Fire Testing Technology, East Grinstead, UK); samples with dimensions of 100 × 100 × 3.2 mm^3^ were exposed to a radiant cone at 35 kW/m^2^ heat flux according to ISO 5660. The fire resistance of the coating was analyzed by the cabinet method test, carried out with reference to the appendix B of standard GB 12441-2018. The weight loss and char index (multiplication of the maximum length, width, and depth of the charring in the wood substrates) of the specimens were measured; the sample size was 300 × 100 × 3 mm^3^. An NBS smoke density box (FTT0064, Fire Testing Technology, UK) was used to test the smoke density D_s_ of the material according to the standard ISO 5659-2. The thermal radiation power was 50 kW, the test mode was flame mode, and the test time was 10 min. The maximum smoke density D_m_ (i.e., maximum specific optical density) was recorded. The material sample was 75 × 75 × 1 mm^3^.

The chemical structure of the char residue was determined with Fourier transform infrared spectroscopy (FT-IR, Nicolet IS50, Thermo fisher, Massachusetts, USA) using a KBr disk, and the wavenumber was set from 4000 cm^−1^ to 410 cm^−1^. The composition was characterized with X-ray photoelectron spectroscopy (XPS, XSAM80 Kratos Co, Manchester, UK). Dielectric properties were analyzed with a broadband dielectric spectrometer (Concept 50, Novocontrol GmbH Corporation, Montabaur, Germany). Raman spectra were recorded with a DXR2xi Raman Spectrometer (Thermo fisher, Massachusetts, USA).

## 3. Results and Discussion

### 3.1. Morphology Characterization and Menchanical Preoperties

To study the degree of modification of the wood by the treatment agents, the weight gain of each treated sample was recorded. As shown in Figure 1b, the weight gain in RW was 2.36%, indicating that CBP was successfully impregnated into the wood. XTW and CXTW exhibited high weight gains of 16.93% and 19.11%, respectively, due to the fixation of the fire-retardant coating on the surface of the wood. The microscopic morphology and element distribution of the wood cross sections were observed using SEM and EDX images. As seen in Figure 1c, RW showed a regular pore structure with pore sizes in the order of tens of microns, and its elemental composition was mainly C and O elements, as observed by EDX. The CTW exhibited a microscopic pore structure similar to that of the RW, and the characteristic elements of CBP (B and Cu) were evenly distributed in the wood, indicating that the preservative penetrated the CTW well (Figure 1d). The cross-section images of XTW and CXTW are shown in Figure 1e,f. A similar connection between the coating and the wood can be observed in each. In addition, the X-1 fire retardant coating formed a dense layer with a thickness of approximately 80 μm after curing, and there was no obvious interface between the coating and the wood, confirming a tight bond between them.

To illustrate the stability of the coating on the wood, we evaluated the adhesion of the coating. Shown in Figure 2b, the test specimens had tensile strengths of 1.83 ± 0.28 MPa, and it is worth noting that cohesive damage occurred in the wood matrix rather than the wood/coating interface when the test specimens were stretched to failure. In other words, the coating remained firmly bonded to the surface of the wood until the test specimen was broken, indicating the coating’s stable adhesion. To evaluate the mechanical properties of the different samples, we conducted compression and bending tests. The results obtained for the compression and bend strengths of the samples are shown in Figure 2c. The compression strengths of RW, CTW, XTW, and CXTW were 45.5, 42.8, 48.9, and 46.3 MPa, respectively. The compression strength of CTW was similar to that of RW, indicating that the preservative treatment had little effect on the mechanical properties of the wood. The higher strength of the wood treated with the fire-retardant coating compared to RW may have been caused by the cured coating on the surface of the wood. Similar results were found in the bending strength data, where CXTW exhibited the highest bending strength (83.3 MPa) among the samples.

### 3.2. Thermal Decomposition Behavior

Thermogravimetric analysis (TGA) was implemented to analyze the thermal decomposition behavior and thermal stability of the samples under a nitrogen atmosphere. The TGA and derivative thermogravimetric (DTG) curves are shown in Figure 3, and the relevant parameters are summarized in Table 1. The initial decomposition temperature at 5 wt% weight loss (*T_5%_*) and the maximum decomposition temperature (*T_max_*) of RW were 281 °C and 387 °C, respectively. According to the reference and DTG curves, the thermal decomposition processes of RW consisted of three main steps [17,18]. The first step was principally due to partial degradation of hemicellulose and lignin into char residues, CO_2_, CO, CH_4_, CH_3_OH, CH_3_COOH, etc. Previous research demonstrated that the depolymerization of hemicellulose occurs between 180 and 350 °C, random cleavage of the glycosidic linkage of cellulose occurs between 275 and 350 °C, and degradation of lignin occurs between 250 and 500 °C [19]. For RW, the degradation of hemicellulose takes place at approximately 340 °C, where a slight shoulder in the DTG curve can be seen in Figure 3b. The second step was mainly caused by the continuous degradation of lignin and cellulose decomposition, during which the less stable aliphatic groups were preferentially decomposed through homolytic cleavage of C–C and C–H bonds; the resultant product is a highly condensed and cross-linked carbonaceous chemical. At approximately 350 °C, degradation of cellulose occurred, and a prominent peak appeared at 387 °C. In the third step, the residual wood components continued being aromatized and carbonized, leading to 16.7 wt% char residue. In comparison, the thermal decomposition behavior of CTW was distinctly different from that of RW. The T_5%_ of CTW decreased to 242.5 °C after being treated by CBP, probably due to the Cu^2+^ and borate promoting the bond cleavage and thermal degradation of the main components in the wood, thus leading to earlier initial thermal decomposition. In addition, different from the sharp mass-loss peak for RW shown in the DTG curves, the mass-loss peak of CTW was relatively smooth, and its maximum rate of mass loss (V_max_) decreased by 60.3% compared with RW, showing that the decomposition of the CBP-treated wood tended to be much steadier. The amounts of combustible volatiles decreased while the residue char increased. These results can be attributed to the H_2_BO_3_ decomposing to B_2_O_3_ and forming a glassy protective surface layer [20,21].

XTW also exhibited a lower T_5%_ (247 °C) than RW, due to the earlier pyrolysis of the fire-retardant coating (according to the components of the coating, the released volatiles included NH_3_ and some small segments [22]). After further thermal pyrolysis, the fire-retardant coating formed polyphosphoric acids as an acid source, promoting the charring of wood [8]. Notably, the charred coating acts as a mass and heat transfer barrier, delaying the thermal degradation of XTW so that a lower V_max_ and higher residue are observed (Table 1) [23].

The differences in thermal degradation behavior between CXTW and XTW are mainly in the first and second steps. CXTW reached T_5%_ earlier than XTW, with similar findings for T_max_, attributed to further catalytic carbonization of CBP under the protection of the fire-retardant coating. Moreover, CXTW retained more char residue, which increased by 42.4% compared to XTW, indicating that the stability and amount of char residue of CXTW were improved, which is beneficial for the flame retardancy of wood. These results suggest that treatment with CBP and fire-retardant coating can significantly catalyze wood char formation and improve thermal stability.

### 3.3. Flam- Retardant Properties

The cabinet method is an important means to evaluate the fire protection of fire-retardant coatings under laboratory conditions, by testing weight loss and char index [24]. The images produced during the test and the results of different samples are given in Figure 4. As shown in Figure 4a, RW burned violently with rapid spread under the alcohol flame. After the test, its weight loss and char index were 58.1% and 188.7 cm^3^, respectively, indicating that RW is highly combustible (Figure 4c). The addition of CBP can suppress the burning of wood to a certain degree. CTW was also ignited by the fire source, but its flame spread was less rapid than RW (Figure 4a), showing a lower weight loss (35.8%) and char index (144.0 cm^3^). In comparison, XTW did not ignite during the test, with weight loss (4.1%) and char index (46.3 cm^3^) significantly lower than those of RW, showing excellent fire protection. This is because the fire-retardant coating formed a compact and intumescent char that prevented the underlying substrates from further burning (Figure 4c). CXTW could not be ignited, similar to the phenomenon of XTW. Moreover, it exhibited a lower weight loss of 3.2% and a char index of 42.8 cm^3^, suggesting that the introduction of CBP can contribute to a good cooperative effect on the fire-protection performance of coatings.

A cone calorimeter (CC) was further employed to study the burning behavior of different samples, to characterize their heat release and smoke release. The CC apparatus is one of the most acceptable for bench-scale burning tests, with a high similarity to real fires, is widely used to evaluate the fire resistance of materials. The method involves supplying a uniform heat-radiation environment from a conical heater [25]. The test results are shown in Figure 5 and Table 2. Time to ignition (TTI) is an important parameter for evaluating the fire-resistance performance of materials, referring to the time taken from commencement of heating to the sustained burning of the material surface at a predetermined intensity of incident heat flow. Due to their high flammability, RW and CTW burned in 10 s and 8 s, respectively (Table 2). After the application of the fire-retardant coating, the TTI of XTW was extended to 35 s, demonstrating significant improvement in fire resistance. Notably, further treatment with CBP did not negatively affect the burning behavior of the wood, and CXTW exhibited a TTI of 32 s, maintaining good fire resistance.

The heat release rate (HRR) and peak heat release rate (PHRR) are important parameters for measuring the fire hazard of materials, referring in this case to the rate of heat release per unit area and the maximum heat release rate during the test, respectively. Figure 5a illustrates two heat release peaks in RW, occurring in the flaming combustion stage [26]. At this stage, carbonization occurred in the wood surface after thermal pyrolysis, and combustible gases, including carbon monoxide, methane, etc., were generated. When encountering open flames, these gases evolved into flash ignition and participated in the burning, resulting in the first heat-release peak. As the fire progressed longitudinally toward the bottom of the wood, combustible gases continued to be produced in large quantities, and the wood burned more intensely, resulting in the second heat-release peak [27,28]. ~According to the data in Table 2, RW had a high PHRR of 410 kW/m^2,^ and its fire growth rate (FIGRA) reached 5.46 kW/(m^2^ s), confirming its high flammability. In comparison, CTW produced a much gentler HRR curve with a PHRR of 262 kW/m^2^ and FIGRA of 3.27 kW/(m^2^ s). This result shows that CBP significantly suppressed exothermic burning and can improve the fire safety of wood to a certain extent. Due to the heat-induced intumescent char layers delaying the thermal decomposition of the wood, XTW remarkably suppressed the heat release. The PHRR and FIGRA of XTW were 254 kW/m^2^ and 2.12 kW/(m^2^ s), 38.0% and 61.2% lower than those of RW, respectively, illustrating the XTW-treated sample’s significantly improved fire safety. Importantly, CXTW exhibited the lowest PHRR (152 kW/m^2^) and the lowest FIGRA (1.39 kW/m^2^ s), implying its superior heat suppression and higher fire safety effect compared with the other samples. This excellent heat release suppression capability was even better than that of XTW, which can be explained by the cooperative effect of the CBP and the fireproof coatings.

Total heat release (THR) refers to the total heat released per unit area during the burning process of a material. In Figure 5b, the THR curve of RW can be seen to grow rapidly until reaching 26 MJ/m^2^, indicating that it underwent violent burning and spreading under high thermal radiation. Table 2 shows that the char residue was only 15.72 wt%, proving a quite adequate burning process. The THR (24 MJ/m^2^) of CTW was obviously lower than that of RW, due to Cu^2+^ and borate promoting the conversion of some organic components to stable char structures. However, due to the oxidative degradation of the char layer under sustained high radiation, the content of residual char was not much different from that of RW. Furthermore, the THR of XTW was not reduced as expected, although the HRR was restricted to some extent because the fire-retardant coating exhibited a limited ability to withstand high thermal radiation. Under the influence of continuous high temperature, the original structure of the porous char layer was damaged, resulting in decreased thermal insulation and increased risk of flammability. In contrast, by the cooperative effect of fire-retardant coating and CBP, CXTW formed more stable char layers, thus inhibiting heat release, resulting in the lowest THR value (23 MJ/m^2^) and the highest char residues (36.22 wt%). These results were consistent with the TGA results (Figure 3). Compared with other flame-retardant wood, the CXTW in this study exhibited higher PHRR reduction (Appendix A) [9,17,29,30,31,32,33].

Furthermore, the smoke release of the different samples was investigated by smoke density testing. Smoke which contains soot particles and toxic gases is the most fatal factor in fires. The dense smoke decreases visibility range in the affected space, restricting chances of escape and directly harming evacuees. Thus, the smoke generation level of materials is a major concern in fire safety risk assessment. In Figure 6, the smoke-density curve for RW can be seen to rise immediately upon ignition and level off as the wood continued to burn, reaching the maximum smoke density (Ds_max_, 86) at approximately 180 s. A low smoke release was found for CTW, with a Ds_max_ of 18.4 which was only about 21% that of RW, indicating that the CBP suppressed the smoke release of the wood. For XTW, the intumescent char layer acted as a physical barrier to smoke release, leading to the smoke-density curve increasing slowly. However, the curve can be seen to rise rapidly after 100 s, eventually reaching Ds_max_ of approximately 80, similar to that of RW, indicating that the fire-retardant coating made almost no contribution to the suppression of smoke release during the burning process. CXTW exhibited a lower Ds_max_ (60.6) than XTW, suggesting that smoke release behavior can be effectively improved by further introduction of the CBP component into fire retardant-treated wood.

### 3.4. Char Residue Analysis

The mechanism of intumescent flame retardation in APP-based fire-retardant coatings having already been well established, our study focused on the contribution of CBP to flame retardancy in wood, and its cooperative effect with fire-retardant coatings on the surface of wood. We therefore removed the intumescent char formed by the fire-retardant coating before char residue analysis, to exclude its effects.

The chemical components of the char residues were characterized by FTIR and XPS. The FTIR spectra of the char residues for each sample are shown in Figure 7. For RW, three intense bands for char residues can be observed in the range 900–700 cm^−1^, owing to aromatic semicircle stretching, indicating the formation of an aromatic structure. The bands at 3000–2800 cm^−1^ corresponding to the symmetrical stretching vibration -CH_2_ and asymmetrical stretching vibration -CH_3_ were weak, which is consistent with the carbonization of holocellulose during burning [34]. In addition, peaks between 3700 and 3100 cm^−1^ were detected in relation to the O-H stretching mode in the -OH groups. For CTW, characteristic bands of B_2_O_3_ are observed at 1353 and 1030 cm^−1^ associated with B-O stretching vibrations, with transmission peaks at 750 and 700 cm^−1^ generated by bending vibrations of the B-O-B bond and the peak at 576 cm^−1^ corresponding to B-O bond wobble vibrations. In the FTIR spectrum of XTW, the strong absorption peak near 491 cm^−1^ represents a special peak of O=P-OH, and the change in this peak from absent to present indicates that the thermal decomposition process of APP produced phosphoric acid. Similar to CTW and XTW, the CXTW sample exhibited a relative abundance of functional groups in the wood after burning.

To further explore the mode of flame-retardance action of CBP in wood, the element contents of the residual char from the smoke density tests were analyzed via XPS. Unlike the char layers of RW containing only C and O elements, the char layers of CTW were composed of C, O, and B elements, as observed in the full-scan XPS spectrum (Figure 8a). The boron ratios in the residue were relatively high, indicating that most of the boron compounds were reserved in the char residue and worked mainly in the condensed phase. Figure 8b shows the results of curve fitting into two Gaussian components for the B1s of CTW. The peaks at 192.4 and 193.2 eV can be assigned to the B-O bonds and B-O-C bonds, respectively, and the relative content of B-O-C bonds was 63.7%, indicating that B was present mainly in the form of crosslinking structures that remained in the char residue.

Small amounts of N and P elements were found in the XTW sample, after the flame-retardant coating penetrated slightly into the surface layer of the wood. As shown in Figure 8a and Table 3, C, O, N, P, and B elements were observed for CXTW. Fire-retardant coatings are known to operate by forming an intumescent char layer and releasing inert gases, while B_2_O_3_ formed in the wood inhibits heat and gas exchange between the wood and the outside world through a condensed-phase flame-retardant mechanism. The element composition of the residue revealed that CBP strengthened the condensed-phase flame-retardant effect of the fire-retardant coating system, which contributed to the cooperative flame-retardant effect during burning.

To further investigate the condensed-phase flame-retardant mode of action, Raman spectroscopy was applied to analyze the char residues. Typically, the integral intensity ratio (I_D_/I_G_) of the D-band (defect, 1365 cm^−1^) and G-band (graphite, 1600 cm^−1^) is used to evaluate the microcrystal size of the char layer [35,36]. Higher I_D_/I_G_ values represent better flame-retardant properties [37,38]. The Raman spectra of the char residue for each sample after smoke-density testing are shown in Figure 9. RW had the lowest I_D_/I_G_, with both CTW and XTW being higher, which indicates that CTW and XTW can form a highly compact char layer after burning. The compact char layer can form a barrier that inhibits mass and heat transfer during combustion, thus inhibiting burning [39]. In addition, CXTW had a slightly lower I_D_/I_G_ value than XTW, which may be attributed to the CBP promoting the formation of a more graphitized structure under fire-retardant coating protection. These results are consistent with our previous inferences.

Based on the aforementioned, we can make the following reasonable assumptions about the flame-retardant and smoke-suppression behavior of CXTW. On the one hand, During the burning process, the fire-retardant coating forms intumescent char layers with a good thermal and matter insulation effect (Figure 10). Subsequently, the intumescent char layer is further oxidized and cracks under thermal radiation, leading to heat being transmitted into the wood. At high temperatures, Cu^2+^ and borate promote the formation of aromatic structures with cellulose and hemicellulose; meanwhile, borate also transforms into B_2_O_3_ as a linkage structure. Furthermore, compact char layers on the wood surface are formed, to inhibit the release of some combustible gases and reduce the heat release. Thus, the flame-retardant coating and the CBP exhibit a cooperative effect in the condensed phase and effectively reduce the heat and smoke release during wood burning.

## 4. Conclusions

This work has revealed the effects of CBP and fire-retardant coatings on the thermal degradation and burning behavior (flame retardancy, heat release, and smoke release) in wood. The results show that XTW exhibited excellent flame retardancy as a consequence of the formation of an intumescent coating, although its smoke release behavior was not ideal. CBP formed a heat and smoke barrier on the surface of the wood by facilitating the charring process at high temperatures, suppressing heat release and reducing smoke release. More importantly, introducing CBP into wood with fire-retardant coating treatment helped to further inhibit heat release and effectively reduced smoke emissions, demonstrating that CBP and fire-retardant coatings can exert strong cooperative effects in improving fire safety in wood. In conclusion, this work reveals the contribution of CBP and fire-retardant coatings to fire retardancy in wood, which is important for improving the fire safety of wood products, and provides guidelines for the further design of reasonable and high-performance fire-retardant wood.

## Figures and Tables

**Figure 1 polymers-14-03944-f001:**
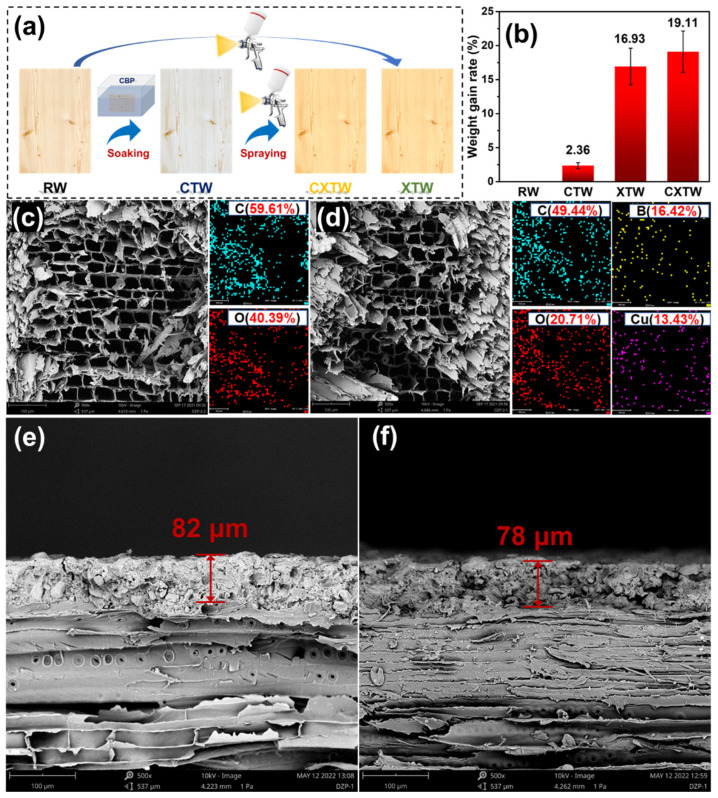
(**a**) Schematic representation of the preparation of different samples; (**b**) weight gain rate of samples; (**c**,**d**) SEM images and EDX mapping results of RW and CTW; (**e**,**f**) SEM images of XTW and CXTW cross sections.

**Figure 2 polymers-14-03944-f002:**
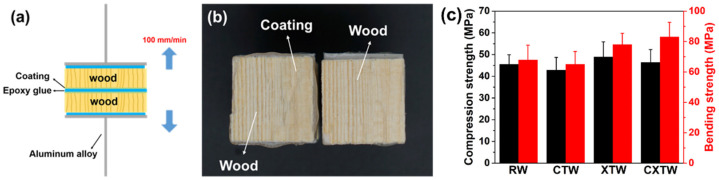
(**a**) Diagram of the adhesion strength test; (**b**) digital photograph of the failed interface after the test; (**c**) compression and bending strength of the samples.

**Figure 3 polymers-14-03944-f003:**
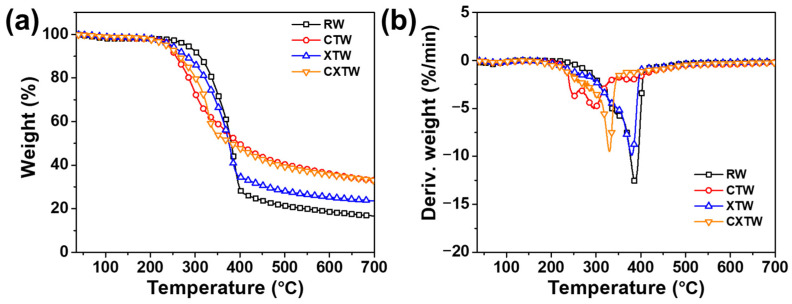
(**a**) TGA and (**b**) DTG curves for RW, CTW, XTW, and CXTW in a nitrogen atmosphere.

**Figure 4 polymers-14-03944-f004:**
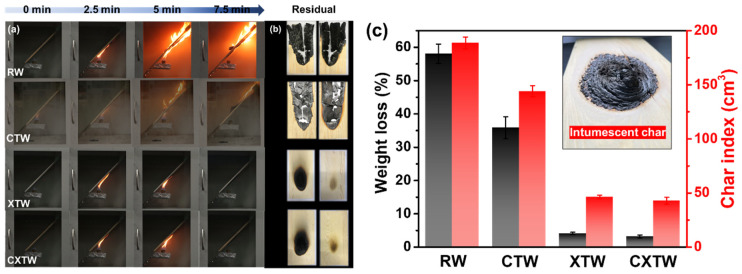
The results of the cabinet test for different samples. (**a**) Burning of RW, CTW, XTW, and CXTW at different times. (**b**) Images of each sample after the test. (**c**) The weight loss, char index, and a digital photograph of the intumescent char formed by fire-retardant coating.

**Figure 5 polymers-14-03944-f005:**
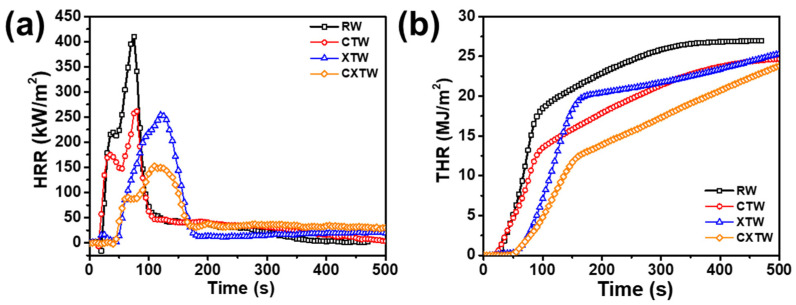
(**a**) HRR, (**b**) THR curves of different samples in cone-calorimeter tests.

**Figure 6 polymers-14-03944-f006:**
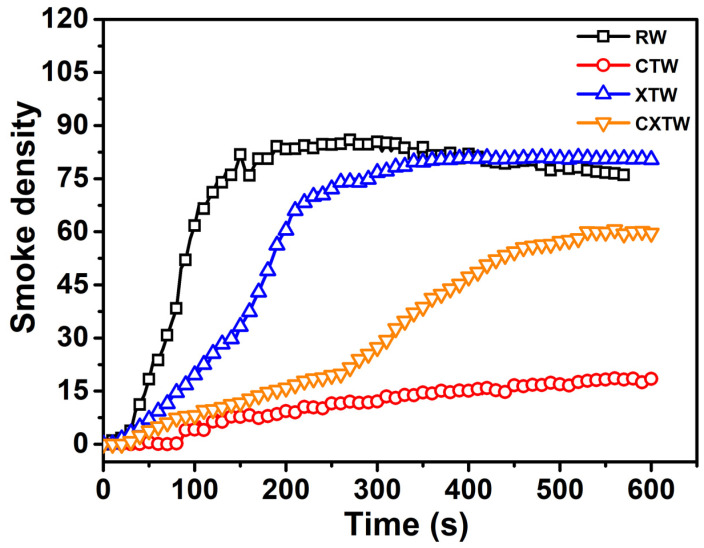
Smoke-density curves of different samples.

**Figure 7 polymers-14-03944-f007:**
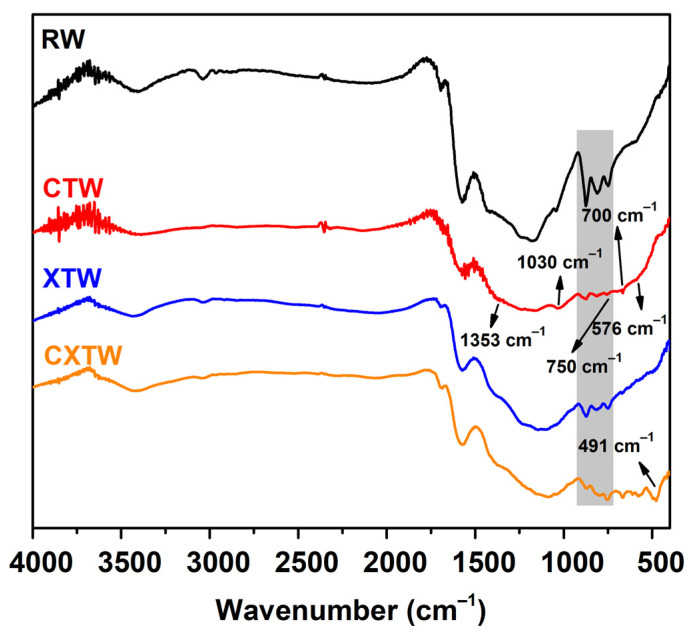
FTIR spectra of the char residues for each sample.

**Figure 8 polymers-14-03944-f008:**
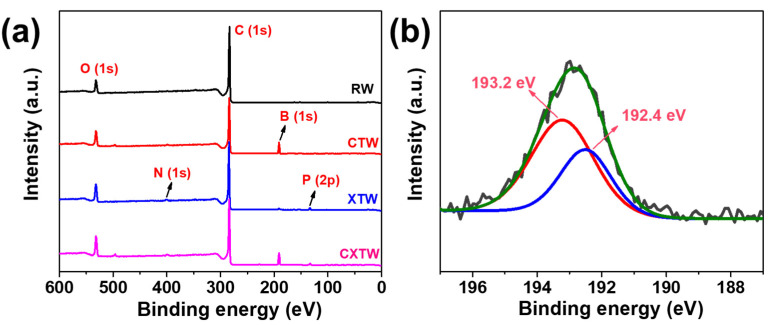
(**a**) Full-scanned XPS spectra of RW, CTW, XTW, and CXTW. (**b**) Deconvolution of B1s spectra of CTW.

**Figure 9 polymers-14-03944-f009:**
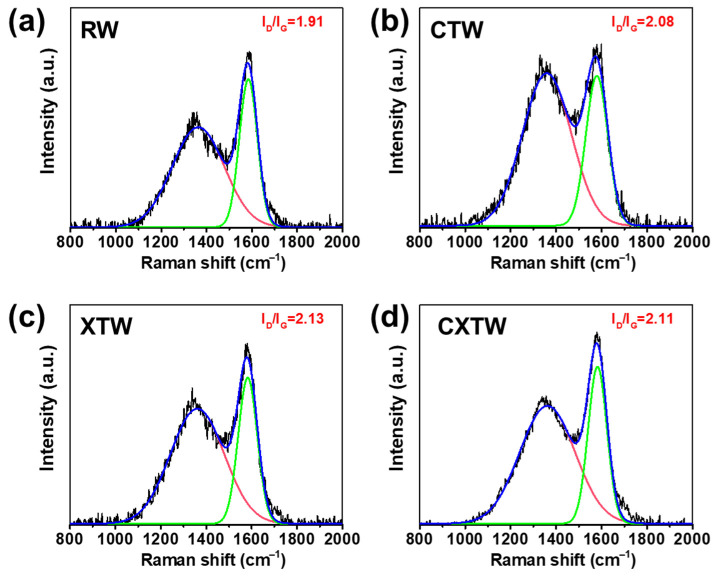
Raman spectra of the (**a**) RW, (**b**) CTW, (**c**) XTW, and (**d**) CXTW residues after smoke-density tests.

**Figure 10 polymers-14-03944-f010:**
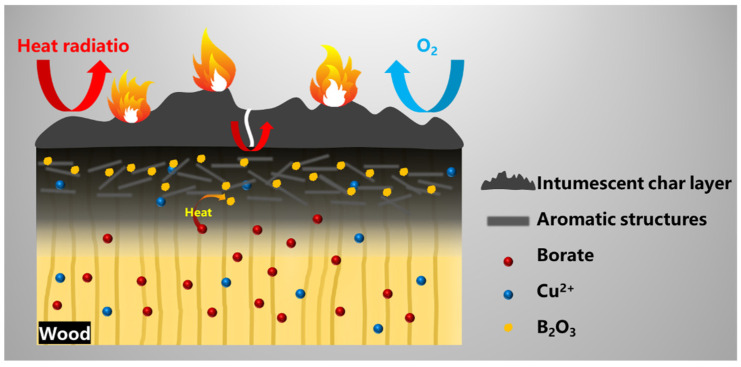
Possible flame-retardant and smoke-suppression mechanisms for CXTW.

**Table 1 polymers-14-03944-t001:** TGA data for different samples in a nitrogen atmosphere.

Sample	T_5%_ (°C)	T_max_ (°C)	V_max_ (%/min)	Residue (%)
RW	281	387	−12.6	16.7
CTW	243	295	−5.0	33.2
XTW	247	380	−9.6	23.6
CXTW	236	331	−9.4	33.6

T_5%_ represents the initial degradation temperature. T_max_ represents the maximum degradation rate, and V_max_ represents the maximum degradation rate.

**Table 2 polymers-14-03944-t002:** Summary of cone-calorimeter and NBS smoke-chamber test results.

Samples	TTI (s)	THR (MJ m^−2^)	PHRR (kW m^−2^)	Residue (wt%)	FIGRA (kW/(m^2^ s))	Ds_max_
RW	10	26	410	15.72	5.46	86.0
CTW	8	24	262	18.38	3.27	18.4
XTW	35	25	254	30.17	2.12	80.4
CXTW	32	23	152	36.22	1.39	60.6

**Table 3 polymers-14-03944-t003:** The element contents of residual char from the smoke-density tests.

Samples	C1s (%)	O1s (%)	B1s (%)	N1s (%)	P2p (%)
RW	91.50	8.50	-	-	-
CTW	74.85	14.75	10.40	-	-
XTW	82.79	12.40	-	2.32	2.49
CXTW	70.80	15.79	9.75	2.20	1.46

## Data Availability

The data presented in this study are available on request from the corresponding author.

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
