# Peer review of "Highly Flame-Retardant and Low Heat/Smoke-Release Wood Materials: Fabrication and Properties"

_polymers, 2022, doi:10.3390/polym14193944_

Round 1
Reviewer 1 Report
The authors have demonstrated the synergistic effect of combining a copper/boron additive (CBCP) to wood, along with a traditional intumescent flame retardant coating. FTIR, XPS, TGA, cone calorimetry, etc. are all used to characterize wood with and without these treatments. It is very clear that CBCP improves the flame retardant behavior of the wood when paired with the intumescent coating. There is good explanation about how the copper and boron act in the condensed phase to promote and strengthen char. There is a lot of useful information here, but there are a few items that need to be addressed before this manuscript is ready for publication:
1. There are many English grammar problems throughout the text. The entire manuscript needs a thorough copyedit to fix all of these problems. This will improve the impact of this nice work.
2. When discussing the cone calorimeter results on page 8, the high char residue of of the CXTW sample is noted as 17.55 wt%. It would be useful to know what wt% the treatments added to the wood. This should be mentioned here to verify how good this result is. In other words, if the treatments are 20 wt%, this result is not very impressive. If the treatments are only 5 wt%, this is a much more significant result.
3. In connection to Point 2 (above), it would be useful to include "weight added" to the samples in either Table 1 or Table 2. This is a key parameter that influences flame retardant behavior and should be clearly presented.
Reviewer 2 Report
The manuscript of Fu et al. deals with investigations on the fire behavior of pine wood samples treated with a preservation agent and equipped with an intumescent flame-retardant coating as well.
The preservation agent is based on boron and copper compounds; the mixture used to deposit the coating contains ingredients normally used for intumescent layers (ammonium polyphosphate, pentaerytrihol, melamine, melamine-urea-formaldehyde resin). However, the authors did not give any additional information in subjection 2.1 “Material”. Therefore, nobody could recapitulate the trials and recheck results of the study. Thus, the authors should specify the contents of the several ingredients of the preservative and intumescent coating, respectively. In addition, the authors do not provide information whether the binder (urea-melamine-formaldehyde resin) was subjected to a curing process. Therefore, it remains unclear, whether the crosslinking process was complete and the obtained coating is stable enough. The authors thoroughly investigated the thermal and flame retardant properties of the wood samples by established methods (TGA, UL94 V test, cone calorimetry, cabinet method test). The char residues remained after burning test were analyzed to gain information on the flame-retardant mode. Pure samples, samples only treated with the preservation or coating agent were investigated for comparison. The investigations revealed best flame retardant properties of samples both treated with the preservative and the intumescent coating. Also important, these specimens showed reduced smoke release in the cone test. The authors argue that there is a synergistic effect of the preservative and the coating. In my opinion, it seems rather to be an additional effect (investigations are necessary to decide whether a synergistic effect is existing or not).
On the whole, the presented manuscript contains interesting findings and should be published after supplementation of subsection 2.1.
Author Response
Dear Editor and Reviewers,
Thanks very much for your valuable comments on our manuscript. The manuscript is carefully revised according to the comments, and the detailed corrections marked in red in the revised version of our manuscript are listed below point by point. We now submit the revised manuscript and look forward to your kindly consideration.
Best Regards,
Teng Fu and Fei Song
-------------------------------------------------------------------------------------------------------------------------------
Reviewer 2:
- The preservation agent is based on boron and copper compounds; the mixture used to deposit the coating contains ingredients normally used for intumescent layers (ammonium polyphosphate, pentaerytrihol, melamine, melamine-urea-formaldehyde resin). However, the authors did not give any additional information in subjection 2.1 “Material”. Therefore, nobody could recapitulate the trials and recheck results of the study. Thus, the authors should specify the contents of the several ingredients of the preservative and intumescent coating, respectively. In addition, the authors do not provide information whether the binder (urea-melamine-formaldehyde resin) was subjected to a curing process. Therefore, it remains unclear, whether the crosslinking process was complete and the obtained coating is stable enough.
Response: We are grateful for your constructive review of our manuscript. According to your useful comments, we have enriched the "Materials and methods" section with detailed information on fire-retardant coatings and preservatives, including the content of the components and the curing time of the coating.
Added section (line 74, page 2): Pinus sylvestris wood was purchased from a wood factory in China; an X-1-type fire retardant coating mainly containing 25 wt% ammonium polyphosphate (APP), 15 wt% pentaerythritol (PER), 10 wt% melamine (MEL) and 50 wt% melamine–urea–formaldehyde resin (MUF), and a copper/boron preservative (CBP) mainly containing Cu2+ and borate (the mass ratio was 1:1) were prepared by the National Engineering Laboratory of Eco-Friendly Polymeric Materials (Sichuan) Lab of Sichuan University.
Added section (line 84, page 2): The above step was repeated 3 times until the total coating amount reached 250 g/m2, then the fireproof and preservative wood are obtained after 8 h curing time.
- The authors thoroughly investigated the thermal and flame retardant properties of the wood samples by established methods (TGA, UL94 V test, cone calorimetry, cabinet method test). The char residues remained after burning test were analyzed to gain information on the flame-retardant mode. Pure samples, samples only treated with the preservation or coating agent were investigated for comparison. The investigations revealed best flame retardant properties of samples both treated with the preservative and the intumescent coating. Also important, these specimens showed reduced smoke release in the cone test. The authors argue that there is a synergistic effect of the preservative and the coating. In my opinion, it seems rather to be an additional effect (investigations are necessary to decide whether a synergistic effect is existing or not).
On the whole, the presented manuscript contains interesting findings and should be published after supplementation of subsection 2.1.
Response: Thank you for your constructive comments. We have revisited our manuscript and continue to believe that there is a synergistic effect of the preservative and the coating. In fact, the synergistic effect can be verified by the two-component action, which exhibits significantly better performance than applying only one component. In this work, the synergistic effect of the preservative and the fire retardant coating is mainly reflected in the highly reduced heat release. As shown in Figure 4 and Table 2, CTW and XTW have PHRRs of 262 kW/m2 and 254 kW/m2, respectively, implying their improved fire safety compared to RW. CXTW exhibits a much lower PHRR (152 kW/m2), and this excellent heat release suppression capability is significantly better than using the preservative or the fire retardant coating alone, which can be strong evidence of the synergistic effect of CBP and fire-retardant coatings.

Reviewer 3 Report
In this manuscript (polymers-1813154) entitled “High flame-retardant and low heat/smoke-release wood materials: Fabrication and properties”, authors have reported the design and preparation of a flame-retardant and smoke-suppression wood based on copper/boron composite preservative and a X-1 type fire-retardant coating. The manuscript is well-organized and the conclusion is supported by the experiment and results. Therefore, this reviewer would suggest an acceptance after addressing the following issues.
1. In introduction section, one recent,important and highly relevant review article should be considered: PAN Mingzhu, DING Chunxiang, ZHANG Shuai, HUANG Yanping. Progress on flame retardancy of wood plastic composites.Journal of Forestry Engineering,2020,5(05):1-12.doi:10.13360/j.issn.2096-1359.202001020
2. In the process of sample preparation, how far is the spray gun from the sample and how long is it sprayed each time? Is the wood sprayed on both sides? What is the chemical structure of CBCP and how to prepare it?
3. As shown in Fig. 1d, e, why is the thickness of CXTW prepared by two-step (impregnating CBCP first and then spraying fireproof layer) smaller than that of XTW prepared by spraying fireproof layer only?
4. Please use the schematic diagram to more intuitively explain the synergy effect between CBCP and fire-retardant coating mentioned many times in the paper. Please explain the condensed phase flame retardant mechanism in combination with samples such as CXTW.
5. Please give the photos before and after the flame retardant tests of the samples to support the argument that "intumescent char" appears many times in the text, because there is no obvious of char intumescent in Fig. 3b and there is no specific explanation in the paper.
6. Please supply relevant videos of flame retardancy tests of RW, CTW, XTW and CXTW.
7.
8. In this paper, authors only compare the flame retardant and smoke suppression properties of the four samples. Please compare the flame retardant properties of CXTW with some other wood-based flame retardant materials. Please carefully read, compare and discuss with other flame retardant materials: LI Xuemin, LIU Yinan, HAO Jianxiu, DUN Mengyuan, WANG Weihong. Flame retardancy of almond shell/high density polyethylene composites. Journal of Forestry Engineering,2020,5(03):80-88. doi:10.13360/ j.issn.2096-1359.201906027; A flame-retardant and transparent wood/polyimide composite with excellent mechanical strength; XU Fucheng, ZHANG Haiyang, LI Yanjun, WU Jianguo. Synergistic flame retardancy of wood treated with zirconium phosphate/ammonium polyphosphate. Journal of Forestry Engineering,2020,5(04):79-86.doi:10.13360/j.issn.2096-1359.201909029
9. There are some grammatical errors in the paper, such as "When the exposure to fire, " on page 1, line 43; “the preservative is penetrated the CTW” on page 3, line 123. In addition, there are some formatting errors in the paper, such as “2h” on page 2, line 84; “3.2mm3” on page 2, line 94. There are extra spaces after "DOI:" on page 13, lines 398 and 404.
10. How about the mechanical properties of the samples? When claim the advantages, especially the mechanical properties of wood, necessary supporting articles should be considered: Mechanical behaviour of wood compressed in radial direction-part I. New method of determining the yield stress of wood on the stress-strain curve; Mechanical behaviour of wood compressed in radial direction: Part II. Influence of temperature and moisture content; etc.
11. Some data in the paper are inconsistent with the corresponding data in the chart. The char residue of RW is 17.0 wt% (page 5, line 151), while it is 16.7% in Table 1. Dsmax of RW is 85 (page 8, line 263), while it is 86 in Table 2. The Dsmax of CTW is 43 (page 8, line 264), which is incorrect and inconsistent with the Fig. 5 and Table 2. There are other similar problems in the paper, please check.
12. “highest FIGRA (1.39 kW/m2 s)” on page 7, line 237 is wrong. In the curve of CTW in Fig. 6, two different peaks are marked as 750 cm-1. The peak of 576 cm-1 mentioned on page 9, line 291 is not seen in Fig. 6. “Fig. 7” on page 10, line 326 should be “Fig. 8”.
13. More comparison with other flame retardant materials should be performed: Metal-Phenolic Network Green Flame Retardants; Low Density, Thermally Stable, and Intrinsic Flame Retardant Poly (bis (benzimidazo) Benzophenanthroline‐dione) Sponge; etc.
14. More details on the used materials should be provided.
15. How about the stability of the coating on the wood? Is the coating easy to be falling out from the composites?
Author Response
Dear Editor and Reviewers,
Thanks very much for your valuable comments on our manuscript. The manuscript is carefully revised according to the comments, and the detailed corrections marked in red in the revised version of our manuscript are listed below point by point. We now submit the revised manuscript and look forward to your kindly consideration.
Best Regards,
Teng Fu and Fei Song
-------------------------------------------------------------------------------------------------------------------------------
Reviewer 3:
- In introduction section, one recent, important and highly relevant review article should be considered: PAN Mingzhu, DING Chunxiang, ZHANG Shuai, HUANG Yanping. Progress on flame retardancy of wood plastic composites.Journal of Forestry Engineering,2020,5(05):1-12.doi:10.13360/j.issn.2096-1359.202001020
Response: Thank you for your careful comment. The introduction to this paper focuses on the flame retardant and preservative work on natural wood, and we do not consider the review on Wood plastic composites (WPCs) to be highly relevant to our work. This is because WPC is a new high performance composite material that is inherently different from natural wood. Meanwhile, due to the complexity of its composition and processing, the combustion process and flame retardant strategies of WPC are also very different from those of wood. Therefore, we decided not to cite this literature.
- In the process of sample preparation, how far is the spray gun from the sample and how long is it sprayed each time? Is the wood sprayed on both sides? What is the chemical structure of CBCP and how to prepare it?
Response: Thank you for your constructive comments. The pneumatic spray gun is 15 cm far from wood surface. We determine the spraying time by recording the weight gain of the wood until the coating density on each side of the wood reaches 250 g/m2. All specimens are sprayed on all surfaces except for the cabinet method test specimens which require only one-sided spraying. We have added these details in lines 83 and 86, page 2.
lines 83, page 2: After drying completely, the CBP-treated wood was painted by X-1 fire retardant coating using a pneumatic spray gun 15 cm from its surface, and the interval between each spraying was controlled at 2 h.
lines 86, page 2: All specimens are sprayed on all surfaces except for the cabinet method test specimens (painting to the surface towards fire only).
Furthermore, we have enriched the "Materials and methods" section with detailed information on fire retardant coatings and preservatives, including the content of the components and the curing time of the coating.
Added section (line 74, page 2): Pinus sylvestris wood was purchased from a wood factory in China; an X-1-type fire retardant coating mainly containing 25 wt% ammonium polyphosphate (APP), 15 wt% pentaerythritol (PER), 10 wt% melamine (MEL) and 50 wt% melamine–urea–formaldehyde resin (MUF), and a copper/boron preservative (CBP) mainly containing Cu2+ and borate (the mass ratio was 1:1) were prepared by the National Engineering Laboratory of Eco-Friendly Polymeric Materials (Sichuan) Lab of Sichuan University.
- As shown in Fig. 1d, e, why is the thickness of CXTW prepared by two-step (impregnating CBCP first and then spraying fireproof layer) smaller than that of XTW prepared by spraying fireproof layer only?
Response: Thank you for your comments. Figures 1e and 1f (corresponding to Figures 1d and 1e in our original manuscript) show the thickness of the fire-retardant coating on the surface of the wood. In the first treatment step (soaked in CBP), the CBP entries into the wood, and does not increase the thickness of the wood. Therefore, the thickness of the coating is only related to the second treatment step (spraying). Notably, we also control the coating thicknesses and the total coating amount of XTW and CXTW to be approximate. The slight difference of thicknesses of XTW and CXTW, shown in the SEM results, is due to errors in manual spraying and SEM characterization.
- Please use the schematic diagram to more intuitively explain the synergy effect between CBCP and fire-retardant coating mentioned many times in the paper. Please explain the condensed phase flame retardant mechanism in combination with samples such as CXTW.
Response: Thank you for your valuable suggestion. We have made a schematic diagram to more intuitively explain the cooperative effect between the CBCP and fire-retardant coating. We have also added this diagram to Fig. 9 in the revised manuscript and mentioned it in the text (line 348, page 11).
Fig. 9 Possible flame retardant and smoke suppression mechanisms for CXTW.
Line 346-348, page 11: On the one hand, the fire retardant coating forms intumescent char layers with a good thermal and matter insulation effect in the burning processes (Fig. 9).
- Please give the photos before and after the flame retardant tests of the samples to support the argument that "intumescent char" appears many times in the text, because there is no obvious of char intumescent in Fig. 3b and there is no specific explanation in the paper.
Response: We could not agree more with your comments. In our revised manuscript, we have added a digital photograph of the intumescent char layer in Figure 3c, page 6 to better support our view.
Figure 3. The results of the cabinet method test of different samples. (a) The burning situation of RW, CTW, XTW and CXTW at different times. (b) Images of each sample after the test. (c) The weight loss, char index and the digital photograph of the intumescent char formed by fire retardant coating.
- Please supply relevant videos of flame retardancy tests of RW, CTW, XTW and CXTW.
Response: Thank you for your suggestion. We have added videos of cabinet method tests on different samples and sent them as supporting information with the manuscript.
- In this paper, authors only compare the flame retardant and smoke suppression properties of the four samples. Please compare the flame retardant properties of CXTW with some other wood-based flame retardant materials. Please carefully read, compare and discuss with other flame retardant materials: LI Xuemin, LIU Yinan, HAO Jianxiu, DUN Mengyuan, WANG Weihong. Flame retardancy of almond shell/high density polyethylene composites. Journal of Forestry Engineering,2020,5(03):80-88. doi:10.13360/ j.issn.2096-1359.201906027; A flame-retardant and transparent wood/polyimide composite with excellent mechanical strength; XU Fucheng, ZHANG Haiyang, LI Yanjun, WU Jianguo. Synergistic flame retardancy of wood treated with zirconium phosphate/ammonium polyphosphate. Journal of Forestry Engineering,2020,5(04):79-86.doi:10.13360/j.issn.2096-1359.201909029
Response: Thank you for your useful suggestion. We have given contrast information on flame retardant properties between this work and other fire retardant treated wood based materials published in recent years in Table R1 below. Compared with other reported flame-retardant wood, the CBP/X-1 coating treated wood in this work has lower PHRR and THR.
Table R1. contrast information on flame retardant properties between this work and other fire retardant treated wood based materials published in recent years.
Materails |
PHRR (kW/m2) |
HRR (MJ/m2) |
TTI (s) |
Ref. |
(PEI/α-ZrP+PEI/APP)10 treated wood |
256 |
74 |
27 |
1 |
Fire-retardant transparent wood |
590 |
27 |
42 |
2 |
Magnesium phosphate ester coated wood |
96 |
3 |
15 |
3 |
AHP-5/MUF treated wood |
147 |
32 |
119 |
4 |
CBP/X-1 coating treated wood |
152 |
23 |
32 |
This work |
Reference:
[1] Xu F, Zhang H, Li Y, et al. Synergistic flame retardancy of wood treated with zirconium phosphate/ammonium polyphosphate. Journal of Forestry Engineering, 2020, 5(04):79-86.
[2] Samanta P, Samanta A, Montanari C, et al. Fire-retardant and transparent wood biocomposite based on commercial thermoset[J]. Composites Part A: Applied Science and Manufacturing, 2022, 156: 106863.
[3] Yan L, Xu Z, Liu D. Synthesis and application of novel magnesium phosphate ester flame retardants for transparent intumescent fire-retardant coatings applied on wood substrates[J]. Progress in Organic Coatings, 2019, 129: 327-337.
[4] Song F, Liu T, Fan Q, et al. Sustainable, high-performance, flame-retardant waterborne wood coatings via phytic acid based green curing agent for melamine-urea-formaldehyde resin[J]. Progress in Organic Coatings, 2022, 162: 106597.
- There are some grammatical errors in the paper, such as "When the exposure to fire, " on page 1, line 43; “the preservative is penetrated the CTW” on page 3, line 123. In addition, there are some formatting errors in the paper, such as “2h” on page 2, line 84; “3.2mm3” on page 2, line 94. There are extra spaces after "DOI:" on page 13, lines 398 and 404.
Response: Many thanks to the reviewers for their responsible and meticulous review work. We have carefully revised the errors in the manuscript.
- How about the mechanical properties of the samples? When claim the advantages, especially the mechanical properties of wood, necessary supporting articles should be considered: Mechanical behaviour of wood compressed in radial direction-part I. New method of determining the yield stress of wood on the stress-strain curve; Mechanical behaviour of wood compressed in radial direction: Part II. Influence of temperature and moisture content; etc.
Response: Thanks for your valuable suggestion. We evaluated the mechanical properties of the different samples by means of compression and bending tests. For bending tests, samples with dimensions of 20 × 20 × 300 mm3 were prepared according to GB/T 1936.1-2009. The compression test samples with dimensions of 30 × 20 × 20 mm3 were prepared according to GB/T1935-2009. The samples were clear of heartwood, knots and visible defects. All samples were oven-dried prior to testing using a temperature of 100 °C.
The bending test was conducted with 15 replicates per sample group on a universal testing machine (INSTRON 3366) equipped with a 10 kN load cell. The span length was set to 240 mm, and the load was applied in the tangential direction at a rate of 10 mm min−1. For compression testing, the load rate of the universal testing machine is 5 mm min−1.
The results of the compression and bending strength of the samples are shown in Fig. R1. The compression strengths of RW, CTW, XTW and CXTW are 45.5, 42.8, 48.9 and 46.3 MPa, respectively. The compression strength of CTW is similar to that of RW, indicating that the preservative treatment has little effect on the mechanical properties of the wood. The higher strength of the wood treated with the fire-retardant coating compared to RW may be due to the cured coating on the surface of the wood. Similar results were found in the bending strength data, where CXTW exhibit the highest bending strength (83.3 MPa) among these samples.
Fig. R1 (a) Compression and bending strength of the samples. (b) The bending stress‒displacement curves of the samples.
- Some data in the paper are inconsistent with the corresponding data in the chart. The char residue of RW is 17.0 wt% (page 5, line 151), while it is 16.7% in Table 1. Dsmax of RW is 85 (page 8, line 263), while it is 86 in Table 2. The Dsmax of CTW is 43 (page 8, line 264), which is incorrect and inconsistent with the Fig. 5 and Table 2. There are other similar problems in the paper, please check.
Response: Thanks for your suggestion. The manuscript has been carefully checked and revised to improve the accuracy.
Line 158, page 5: In the third step, the residual wood components continue to be aromatized and carbonized, leading to 16.7 wt% char residue.
Line 272, page 8: …. reaching the maximum smoke density (Dsmax, 86) at approximately 180 s.
Line 272, page 8: A low smoke release is found in CTW with a Dsmax of 18.4, only about 21% that of RW, ….
- “highest FIGRA (1.39 kW/m2 s)” on page 7, line 237 is wrong. In the curve of CTW in Fig. 6, two different peaks are marked as 750 cm-1. The peak of 576 cm-1 mentioned on page 9, line 291 is not seen in Fig. 6. “Fig. 7” on page 10, line 326 should be “Fig. 8”.
Response: Thank you for pointing out these problems. We have rectified these contents in our revised manuscript.
Line 245, page 7: Notably, CXTW exhibits the lowest PHRR (152 kW/m2) and the lowest FIGRA (1.39 kW/m2 s), ….
Figure 6. FTIR spectra of the char residues for each sample.
- More comparison with other flame retardant materials should be performed: Metal-Phenolic Network Green Flame Retardants; Low Density, Thermally Stable, and Intrinsic Flame Retardant Poly (bis (benzimidazo) Benzophenanthroline‐dione) Sponge; etc.
Response: Thanks for your comments. Although wood and wood-plastic composites, foams and sponges are all building materials, the differences in composition and application make their obvious distinctions in flame retardant strategies and difficulties. A direct comparison between them seems inappropriate. In our manuscript, we have tried to elucidate the flame retardant contribution and synergistic effects of preservatives and fire-resistant coatings in natural wood systems. Therefore, the properties other than flame retardancy and smoke suppression of modified wood are not discussed much, as it is inconsistent with the main theme of the paper. To avoid unnecessary misunderstandings, we have not compared this work with other material.
- More details on the used materials should be provided.
Response: We are grateful for your constructive review of our manuscript. According to your useful comments, we have enriched the "Materials and methods" section with detailed information on fire-retardant coatings and preservatives, including the content of the components and the curing time of the coating.
Added section (line 74, page 2): Pinus sylvestris wood was purchased from a wood factory in China; an X-1-type fire retardant coating mainly containing 25 wt% ammonium polyphosphate (APP), 15 wt% pentaerythritol (PER), 10 wt% melamine (MEL) and 50 wt% melamine–urea–formaldehyde resin (MUF), and a copper/boron preservative (CBP) mainly containing Cu2+ and borate (the mass ratio was 1:1) were prepared by the National Engineering Laboratory of Eco-Friendly Polymeric Materials (Sichuan) Lab of Sichuan University.
- How about the stability of the coating on the wood? Is the coating easy to be falling out from the composites?
Response: We thank the reviewer for the comment. To estimate the adhesion of the coating, strength tests were performed using a universal testing machine (SANS CMT 4204) with coated wood at room temperature. Test specimens were prepared according to the GB/T 14907-2002 standard. The identical coated wooden block (length × width × thickness = 70 mm × 70 mm × 10 mm) was bonded with epoxy resin and immediately covered with another coated wood (Fig. R2a). Then, the samples were compressed with a 1 kg weight for 3 days. After the above step, the samples were pulled to failure at a 100 mm/min rate, and adhesive strength was calculated as the maximum force divided by the known overlap area.
The test specimens showed tensile strengths of 1.83 ± 0.28 MPa, and it is worth noting that damage occurred to the wood rather than the wood/coating interface when the test specimens were stretched to failure (Fig. R2b). In other words, the coating remained firmly bonded to the surface of the wood until the test specimen broke down, indicating that the coating was stable adhesion.
Fig. R2 (a) Diagram of the adhesion strength test of the coating. (b) Digital photograph of the failed interface after the test.

Round 2
Reviewer 3 Report
Authors have responsed all the previous comments. However, necessary revision should be performed in the revised manuscript, expecially the modification of introduction, comparison with previous work, etc. Please carefully make responses to previous comments, and highlight the corresponding revisions point-to0point in the main manuscript.
Round 3
Reviewer 3 Report
Authough authors have made detailed responses to the previous comments, necessary revision according to the comments are highly required in the manuscript. Authors should make corresponding revisions in the main manuscript, especially the comments regarding the comparison with previous work to highlight the nolvety and advantages of their work. Please carefully read the previous comments and make corresponding corrections and discussions in the revised manuscript.
Author Response
Thanks for your attention on our manuscript (Manuscript ID: polymers-1813154) to let us make a minor revisions according to the review report of reviewer 3.However, it seems that the reviewer did not read our response carefully, and still gave the same review reports (Authors should make corresponding revisions in the main manuscript, especially the comments regarding the comparison with previous work to highlight the novelty and advantages of their work). We have not only added the recent, important, and highly relevant article in the introduction but also added the comparison of relevant work in Table S1, which have reflect the novelty and the advantages of our work. Moreover, these response contents have been listed in the response (Round 2). We have made detailed responses to the review comments by point-to-point, and have performed corresponding revisions in our revised manuscript highlighted in red. In the response letter, we also have listed and highlighted the detail page and lines of the revised text to make the reviewer a clear understanding of the changes we have made.Thus, we do not agree with the review comments (Round 3) of reviewer 3.
Sincerely yours,
Teng Fu & Fei Song
Sichuan University
